# Light and Potassium Improve the Quality of *Dendrobium officinale* through Optimizing Transcriptomic and Metabolomic Alteration

**DOI:** 10.3390/molecules27154866

**Published:** 2022-07-29

**Authors:** Yue Jia, Juan Liu, Mengyao Xu, Guihong Chen, Mingpu Tan, Zengxu Xiang

**Affiliations:** 1College of Horticulture, Nanjing Agricultural University, Nanjing 210095, China; 2020104130@stu.njau.edu.cn (Y.J.); 2021804300@stu.njau.edu.cn (M.X.); 2021104131@stu.njau.edu.cn (G.C.); 2College of Life Sciences, Nanjing Agricultural University, Nanjing 210095, China; 2021116018@stu.njau.edu.cn

**Keywords:** light, potassium, metabolomic, transcriptomics, flavonoids, *Dendrobium officinale*

## Abstract

Background: *Dendrobium officinale* is a perennial epiphytic herb in Orchidaceae. Cultivated products are the main alternative for clinical application due to the shortage of wild resources. However, the phenotype and quality of *D. officinale* have changed post-artificial cultivation, and environmental cues such as light, temperature, water, and nutrition supply are the major influencing factors. This study aims to unveil the mechanisms beneath the cultivation-induced variation by analyzing the changes of the metabolome and transcriptome of *D. officinale* seedlings treated with red- blue LED light and potassium fertilizer. Results: After light- and K-treatment, the *D. officinale* pseudobulbs turned purple and the anthocyanin content increased significantly. Through wide-target metabolome analysis, compared with pseudobulbs in the control group (P), the proportion of flavonoids in differentially-accumulated metabolites (DAMs) was 22.4% and 33.5% post light- and K-treatment, respectively. The gene modules coupled to flavonoids were obtained through the coexpression analysis of the light- and K-treated *D. officinale* transcriptome by WGCNA. The KEGG enrichment results of the key modules showed that the DEGs of the *D. officinale* pseudobulb were enriched in phenylpropane biosynthesis, flavonoid biosynthesis, and jasmonic acid (JA) synthesis post-light- and K-treatment. In addition, anthocyanin accumulation was the main contribution to the purple color of pseudobulbs, and the plant hormone JA induced the accumulation of anthocyanins in *D. officinale*. Conclusions: These results suggested that light and potassium affected the accumulation of active compounds in *D. officinale*, and the gene-flavone network analysis emphasizes the key functional genes and regulatory factors for quality improvement in the cultivation of this medicinal plant.

## 1. Introduction

The genus *Dendrobium* is composed of approximately 1500–2000 species living in the tropical and subtropical areas of Asia and North Australia [1,2,3]. Among the *Dendrobium* species, *D. officinale,* one of the most famous species of *Dendrobium*, has long been regarded as a precious herb and health food applied in TCM and in folk, where it is known as “the first of the Chinese nine fairy herbs” [4].

Updated pharmacological research has indicated that *D. officinale* has the functions of enhancing immunity [5,6], anti-aging [7], anti-tumor [8,9], reducing blood sugar [10,11], and has a great curative effect in the treatment of throat diseases, gastrointestinal diseases [12,13], cardiovascular diseases [14], etc. According to currently available phytochemical investigations, *D. officinale* contains various compounds, such as polysaccharides, bibenzyls, phenanthrenes, phenylpropanoids, flavonoids, and alkaloids [15,16]. Among them, polysaccharides, alkaloids, and flavonoids are considered to be the main biologically-active compounds responsible for various pharmacological properties and therapeutic efficacy of *D. officinale* [17].

Transcriptomes can efficiently and comprehensively excavate the gene expression, structure, function, and regulatory mechanism of genes, while metabolomics can detect metabolites that directly affect the metabolic state of the organism or cell during a specific period [18,19]. The release of the genome sequence of *D. officinale* provided a basis for deciphering the molecular mechanism of the biological morphology and active ingredient regulation of *D. officinale* [4,20]. In recent years, some progression based on *D. officinale* transcriptomes and metabolomes has been explored in *D. officinale* polysaccharide [21,22,23], alkaloid [24,25] biosynthesis and accumulation mechanism, seed germination [26], flower development [27,28] and other growth and development regulation, seedling cold acclimation response [29], salt stress response of leaf tissue [30], cadmium stress detoxification mechanism [31], and other abiotic stress response, as well as differences in metabolites in different organs and growth and development stages [32,33]. However, there are few studies on the regulation of flavonoid synthesis. Moreover, there is variation in the stem color during the cultivation of *D. officinale*, with purple stems having better quality and more accumulation of flavonoids [34,35]. Therefore, it is necessary to further investigate the biosynthesis and accumulation of flavonoids in *D. officinale* and their interaction with the environment.

Secondary metabolites generated by medicinal plants are the material basis for their clinically curative effects, and they are also important indicators for evaluating the quality of herbs [36,37]. However, the synthesis and accumulation of secondary metabolites are very complex and are affected by many factors including internal developmental genetic circuits (regulatory genes, enzymes) and external environment factors (light, temperature, water, mineral elements, etc.). Regarding the medicinal herb *Perilla frutescens*, a low nitrogen supply increased flavonoids in leaves and stems, whereas a low potassium or magnesium supply slightly increased rosmarinic acid and certain flavonoids in this medicinal herb through non-targeted metabolite profiling [38]. Potassium, recognized as a “quality element”, is one of the most important nutrient elements affecting the quality and yield of medicinal plants [39]. Potassium can activate plant asparaginase, which plays a crucial role in plant nitrogen mobilization [40]. Under high potassium levels, the activity of phenylalanine ammonialyase increased, and the content of total phenols and flavonoids increased correspondingly in the Malaysian Herb *Labisia pumila* Benth [41]. In addition, different light affects the anabolism of phenylpropane and flavonoids [42]. The phenolic synthesis and free radical scavenging activities of green (acyanic) basil were improved under red light, and, coincidently, the same effects were observed in the red (cyanic) cultivar when exposed to blue light. Moreover, the biosynthesis of rosmarinic and gallic acid was enhanced post-blue illumination [43].

To investigate the effects of mineral elements and light on the synthesis of bioactive metabolites in *D. officinale* pseudobulb, an integrated wide-target metabolomic and transcriptomic analysis was conducted. Particularly, the fluctuation of flavonoid biosynthesis genes and the regulatory network mediated by transcription factors in *D. officinale* post-light- and K-treatment were further analyzed. This study not only deciphers the mechanism of flavonoid synthesis and regulation in *D. officinale* but also provides a theoretical basis for the application of LED light and potassic fertilizer in the high-quality cultivation of *D. officinale*.

## 2. Results

### 2.1. Phenotypic Changes in Light- or K-Treated D. officinale Pseudobulb

*D. officinale* shows differences in biological characteristics such as color, size, hardness, and thickness affected by germplasm and ecological environment factors. From the perspective of environmental factors affecting the morphology and quality of *D. officinale*, we used the one-year seedlings of *D. officinale* with consistent growth status as the material, the pseudobulbs (P) were cultured under natural light or treated with 3 mM KCl (PK), while those treated by red and blue LED light were denoted as PL. After light- and K-treatment, the *D. officinale* pseudobulbs turned purple, and the anthocyanin content in *D. officinale* pseudobulbs increased from 7.48 to 11.74 and to 12.65 μg·g^−1^ post light- and K-treatment, respectively (*p* < 0.05) (Figure 1).

### 2.2. Metabolomic Changes in of Light- or K-Treated D. officinale Pseudobulb

To further explore the impact of light quality and potassium on the metabolic shifts of *D. officinale*, the total extracts were subjected to UPLC-TQMS for nontargeted metabolomics analysis. The significantly differentially accumulated metabolites (DAMs) between sample groups were defined by |FC| > 1.5 and VIP > 0.7 post light- or K-treatment.

After quality validation, compared with the control pseudobulbs (P), 348 DAMs were identified in light-treated pseudobulb PL samples, among which 189 DAMs were upregulated and 159 were downregulated. For the 346 DAMs identified in K-treated pseudobulbs PK samples, 209 were upregulated and 137 were downregulated.

Regarding the DAMs category, alkaloids, phenolic acids, flavonoids, and lignin were more prominent post-treatment; moreover, flavonoids accounted for the highest proportion of metabolites, which was 22.4% and 33.5% in light- or K-treated *D. officinale*, respectively (Figure 2). After light- and K-treatment, there were 39 flavonoids with the same upward and downward trend (Table 1). Among them, naringenin chalcone, pinocembrin, butin, and hesperetin decreased after light- and K-treatment whereas the majority of flavonoid glycosides increased, and these upregulated flavonoid glycosides were mainly quercetin and apigenin derivatives. On the other hand, compared with the control pseudobulb, the light- and K-treatment affected the accumulation of anthocyanins in *D. officinale* pseudobulbs. For example, delphinidin and peonidin were significantly upregulated after light- and K-treatment, indicating that the accumulation of anthocyanins was the main reason for the purple appearance of *D. officinale* pseudobulbs. Interestingly, the phytohormone JA and its derivatives, such as JA, (-)-jasmonoyl-L-isoleucine (JA-Ile), 5′-glucosyloxyjasmanic acid, dihydrojasmone, and cis-jasmone, were significantly upregulated after light- or K-treatment, respectively (Table 1). These results showed that light- and K-treatment affected the biosynthesis and accumulation of primary and secondary metabolites in the stem of *D. officinale*, especially the flavonoids.

### 2.3. Transcriptomic Changes in Light- or K-Treated D. officinale Pseudobulb

To further elucidate the light and K imposed profound impacts on the metabolic alterations, transcriptomic profiling based on RNA-seq was performed to identify differentially expressed genes (DEGs) in *D. officinale*. Using a stringent cutoff (|FC| > 1.5 and *p*-value < 0.1), a total of 3401 DEGs were identified as being light-responsive genes, of which 1832 were light-induced and 1569 were light-repressed, and 2892 DEGs were identified as being K-responsive genes, among which 1173 were K-induced and 1719 were K-repressed in *D. officinale*. Furthermore, nine randomly selected DEGs were verified by RT-qPCR; the results showed that the expression trend of DEGs detected by RT-qPCR was consistent with the transcriptome sequencing data, which confirmed that the RNA-seq transcriptional sequencing results were reliable, and the expression pattern of key genes in the flavonoid biosynthesis pathway, such as *ANS* and *F3H*, was consistent between these two approaches (Appendix A).

To further evaluate the functions and the biological pathways represented by the DEGs, these genes were aligned with those in the KEGG database. After enrichment analysis, the top 20 pathways with the most significant *p*-value enrichment were selected for comparison, and the results showed that light- or K-aroused DEGs were enriched in the biosynthesis-related pathways of flavonoids, such as the “Flavonoid biosynthesis”, “Phenylpropanoid biosynthesis”, and “flavone and flavonol biosynthesis”. These indicated that the genes in the flavonoid biosynthesis pathway were particularly affected by light- and K-treatment (Appendix A).

### 2.4. Identification of Key Genes Correlated to Flavonoid Biosynthesis via Coexpression Analysis

To determine the potential genes regulating flavonoid biosynthesis, gene coexpression analysis was performed to gain insight into the molecular mechanism of the potential association between flavonoids and genes regulated by light and potassium. In total, 14,465 genes were subjected to WGCNA, and 21 coexpression modules (represented by various colors) were identified as a consequence (Figure 3A). To explore the relationship between these modules and the metabolic traits, the correlation of the expression of the genes in these modules with metabolites was evaluated (Figure 3B). It should be noted that the gene expression in the module aliceblue (973 genes) was negatively correlated with the downregulated flavonoids, but positively correlated with the upregulated flavonoid glycosides. Similar change patterns were also observed in the modules antiquewhite 4 (973 genes), green 2 (656 genes), and deeppink 2 (446 genes). In contrast, modules chocolate 3 (1131 genes), mediumpurple (1633 genes), coral 4 (103 genes), and cornflowerblue (303 genes) showed the opposite change patterns compared to module aliceblue. In addition, the phythohormone JA and JA-Ile were significantly positively correlated with genes in module aliceblue, but negatively correlated with those in mediumpurple and cornflowerblue modules. The KEGG enrichment analysis was conducted to further determine the significant regulatory pathways in the aforementioned eight modules (Appendix A). Consequently, the flavonoid biosynthesis pathway was enriched in the aliceblue, deeppink 2, and mediumpurple modules, while the phenylpropane biosynthesis pathway was enriched in the chocolate 3, green 2, and cornflowerblue modules. Moreover, plant hormone transduction was enriched in these six modules. Therefore, DEGs in these six modules enriched in phenylpropane biosynthesis, flavonoid biosynthesis, and linolenic acid metabolism pathway were selected for further analysis (Table 2). Intriguingly, the expression of genes related to flavonoid biosynthesis was upregulated after light- or K-treatment, and these DEGs for flavonoid biosynthesis included *CHS8*, *F3H*, *F3′5′H*, *DFR*, *ANS*, *BZ1*, and *UFGT1*. Moreover, the expressions of the genes in the linolenic acid metabolic pathway for JA biosynthesis were upregulated in K-treated *D. officinale*, as was the case for light treatment.

### 2.5. Gene-Metabolite Correlations Reveal Significant Interactions between Flavonoid-Related DEGs and DAMs

To integrate the metabolome and transcriptome analysis of culture response, we performed canonical correlation analysis using the Pearson correlation coefficient (PCC) to show the dynamic changes during light- and K-treatment. Firstly, to systematically understand the relationship between flavonoid biosynthesis and phytohormone JA, we constructed a network map of DEGs enriched in four KEGG pathways (phenylpropane biosynthesis, flavonoid biosynthesis, linolenic acid metabolic pathway biosynthesis, and plant hormone transmission, with |coefficient| > 0.7) (Figure 4). The network map indicated that the genes for JA synthesis (*AOC3*, *AOS2*, *AOS2L*, *ARR9*, *PER51,* and *TIFY6B*) also were involved in the synthesis of flavonoids.

Regarding the differentially expressed Transcription factors (TFs), 59 TFs in 12 families such as AP2/ERF, bHLH, MYB, WRKY, NAC, and GATA displayed the same changed pattern post light- and K-treatment (Appendix A). To obtain the regulatory relationship between flavonoid biosynthesis and culture-responsive TFs, a network map of 59 transcription factors and 24 flavonoids was constructed (Figure 5). Among them, *ERF039*, *ERF110,* and *MYB44L* were positively correlated with most of the upregulated flavonoid glycosides in culture, whereas *bHLH112* and *bZIP6* were negatively correlated with most of the upregulated flavonoid glycosides in culture.

### 2.6. The Effect of JA on the Synthesis of Anthocyanins in D. officinale Pseudobulbs

Anthocyanins are not only flavonoids, but also plant pigments that affect the apparent color of plants. Through the cross-sectional microscopic observation of *D. officinale* pseudobulbs, it was found that anthocyanins affecting the stem color of *D. officinale* were mainly distributed in the peripheral part of the stem (Figure 6). Compared with the control, exogenous MeJA treatment of *D. officinale* seedlings significantly promoted the synthesis of anthocyanins in pseudobulbs. After 7 days of treatment, anthocyanin accumulation increased significantly, and most of the peripheral cells became pink, but the color was lighter. After 20 days of treatment, anthocyanin accumulation continued to increase, and it was found that the anthocyanin accumulation was not only reflected in the increase in the number of purple cells, but also in the anthocyanin content of single cells. By contrast, the JA inhibitor SHAM treatment inhibited anthocyanin synthesis, after 20 days of treatment, the number of purple cells and anthocyanin accumulation decreased significantly (*p* < 0.05).

## 3. Discussion

### 3.1. Light and K Regulate the Biosynthesis of Flavonoids in D. officinale

The secondary metabolites of medicinal plants are pivotal sources of natural bioactive components, which are closely related to plant growth and environmental factors. Firstly, as a key environmental factor, light affects the accumulation of secondary metabolites. Photon radiation with different wavelengths and intensities is a basic abiotic component required for plant photosynthesis, growth, and the accumulation of secondary metabolites [44,45,46]. Compared with white light, the combination of light with different colors increases the accumulation of diverse primary and secondary metabolites, such as amino acids, carotenoids, tocopherols, soluble sugars, and flavonoids [47,48,49], while potassium is an essential nutrient element affecting most biochemical and physiological processes of plant growth and metabolism, and plays an important role in enzyme activation, photosynthesis, and metabolic processes [50,51,52]. The importance of potassium fertilizer to crop yield and quality formation is well known. In this study, there were differences in the contents of sugar, amino acids, flavonoid alkaloids, phenolic acids, and lignin in light- or K-treated *D. officinale*, among which flavonoids accounted for the largest proportion. In medicinal plants, flavonoids are important secondary metabolites. They play critical roles in many physiological processes such as plant antioxidant and UV-B radiation, auxin distribution and transport regulation, pollen recognition regulation, and so on [53,54]. Moreover, they also play a pharmacological role as a major source of active substances [55,56]. Herein, the transcriptomic analysis showed that in the pathway of regulating flavonoids, many genes showed significant upregulation after cultivation, including *CHS8*, *F3H*, and *F3′5′H1* (Table 2). These significantly upregulated genes are involved in the accumulation of flavonoids, which may also contribute to the difference in flavonoid content in the pseudobulbs of *D. officinale* regulated by light and potassium.

### 3.2. JA Promoted the Synthesis of Anthocyanin Which Contributed for the Purple Stem of D. officinale

Different geographical environments and climatic conditions lead to the resource diversity of *D. officinale*, which shows great differences in the stem, node, leaf, flower, and other characters. According to the color of pseudobulbs, *D. officinale* can be divided into green stems and purple stems. In this study, it was found that after light- and K-treatment, the *D. officinale* pseudobulbs turn purple, the anthocyanin content increased significantly (Figure 1), and the contents of delphinidin derivatives increased significantly (Table 1). The enzymes closely related to anthocyanin synthesis, such as *ANS*, *CYP75B3*, and *BZ1*, were upregulated by light- and K-treatment (Table 2). It can be speculated that the purple appearance of pseudobulbs is caused by anthocyanin accumulation, mainly the diglycosides or glycoside derivatives of delphinidin and anthocyanin. In the *D. officinale* anthocyanin biosynthesis pathway, *ANS* and *BZ1* encoding anthocyanin synthase and anthocyanin-3-o-glucosidase, respectively, are key regulatory genes related to the differential accumulation of anthocyanins.

Plant hormones play an important role in anthocyanin accumulation. Current evidence shows that in Arabidopsis, GA, JA, and ABA regulate the expression of sucrose-induced anthocyanin biosynthesis genes [57]. In strawberry, IAA participates in the development of fruit color by regulating anthocyanin biosynthesis, as well ABA and MeJA can promote the accumulation of anthocyanins [58,59]. In this study, JA and its derivatives JA-Ile were upregulated after light- and K-treatment, and the expression of *AOS2*, *AOS2L,* and *AOC3* encoding the key enzymes of JA biosynthesis in *D. officinale* was coordinately upregulated. Intriguingly, these enzyme genes for JA synthesis were associated with flavonoid glycosides in *D. officinale*. Moreover, JA could induce whereas its inhibitor SHAM suppressed the biosynthesis of anthocyanins in *D. officinale*, which confirmed the relationship between JA and anthocyanins in *D. officinale*. In addition, transcription factors regulate anthocyanin biosynthesis in *D. officinale*, and this was reminiscent of the transcription regulators of the anthocyanin biosynthesis pathway, including R2R3-MYB protein, bHLH protein, and WD protein [60,61]. In this study, the differential expression of AP2/ERF, bHLH, and MYB was identified (Appendix A). In mulberry, *bHLH3* is the core factor of the whole flavonoid homeostasis regulation network. It combines with *MYBA*, *TT2L1*, *TT2L2,* and *TTG1* to form the corresponding MYB-bHLH-WD40 transcription complex, which is involved in activating the synthesis of anthocyanins and procyanidins. The abnormal expression of bHLH3 changes the metabolic flux of the flavonoid pathway in light-colored mulberry, resulting in different contents and proportions of anthocyanins, flavonoids, and flavonols in mulberry, making mulberry have different color characteristics [62]. Therefore, it is necessary to unveil the relationship between JA-TFs-anthocyanin biosynthesis key genes in the future.

## 4. Materials and Methods

### 4.1. Plant Materials and Treatments

The one-year seedlings of *D. officinale* Kimura et Migo were planted in the flowerpot containing a 2:1 mixture of perlite and vermiculite. The pseudobulbs (P) were cultured under natural light or treated with 3 mM KCl (PK). The other group PL were treated with red and blue mixed light for 10 h every day (R:B = 5:1 for the PPFD (photosynthetic photon flux density) of 100 μmol·m^−2^·s^−1^). About 10 pseudobulbs were harvested 2 months post each treatment as one biological replicate, and three biological replicates were sampled, frozen in liquid nitrogen, and stored at −80 °C for the subsequent metabolomic and transcriptomic analysis.

The one-year seedlings of *D. officinale* were planted in the flowerpot containing a 2:1 mixture of pearlite and vermiculite. One cluster is planted in each pot, and the number of plants in each cluster is basically the same. After one week of transplant seedling, 100 μM methyl jasmonate (MeJA) or 500 μM JA inhibitor salicylhydroxamic acid (SHAM) was sprayed on the leaves daily for one week. About 10 pseudobulbs of *D. officinale* were sampled with triplicates on 7 and 20 days of treatment, frozen in liquid nitrogen, and stored at −80 °C for subsequent experiments.

### 4.2. Metabolome Analyses

The widely targeted metabolomic profiling was carried out by Metware Biotechnology Co., Ltd. (Wuhan, China) (www.metware.cn, accessed on 20 September 2021). Briefly, 100 mg of pulverized lyophilized sample powder was dissolved in 1.2 mL 70% methanol solution, vortexed for 30 s every 30 min six times, and the sample was placed in a 4 °C refrigerator overnight. After the centrifugation at 12,000 g for 10 min, the extracts were filtrated (SCAA-104, 0.22 μm pore size; ANPEL, Shanghai, China, www.anpel.com.cn, accessed on 30 September 2021), then analyzed using a UPLC-ESI-MS/MS system (UPLC, SHIMADZU Nexera X2, www.shimadzu.com.cn, accessed on 30 September 2021; MS, Applied Biosystems 4500 Q TRAP, www.thermofisher.cn/cn/zh/home/brands/applied-biosystems.html, accessed on 30 September 2021) [63,64].

The mass spectra of compounds were compared with the existing mass spectrometry databases (NIST 08) and MWDB V2.0 (Metware Biotechnology Co., Ltd. Wuhan, China)) [65]. The quantifications of metabolites were performed using a scheduled multiple reaction monitoring (MRM) method [66], following the data evaluation (PCA, HCA, and PCC analysis), the differentially-accumulated metabolites were screened [67], and then the identified metabolites were annotated using the Kyoto Encyclopedia of Genes and Genomes (KEGG) Pathway database (www.kegg.jp/kegg/pathway.html, accessed on 30 October 2021). Significantly, DAMs between groups were defined by |Fold Change| > 1.5 and Variable Importance in the Projection VIP > 0.7 with *p*-value < 0.05.

### 4.3. RNA Extraction, Library Construction, and Sequencing

Total RNA was isolated using the Trizol Reagent (Invitrogen Life Technologies, Shanghai, China), and 3 μg RNA was used as input material for the RNA sample preparations. The total RNA was quantified, and the quality was assessed using an Agilent 2100 Bioanalyzer system (Agilent Technologies, Palo Alto, CA, USA) [68]. The sequencing library was sequenced on NovaSeq 6000 platform (Illumina) by Personal Biotechnology Cp., Ltd. (Shanghai, China).

### 4.4. Transcriptome Analyses

Raw data were first processed to obtain clean data by removing connectors and low-quality reads. The filtered reads were mapped to the reference genome [4,20] (https://www.ncbi.nlm.nih.gov/assembly/GCF_001605985.2, accessed on 12 December 2021) using HISAT2 (v2.0.5, Johns Hopkins University, Baltimore, USA) and used HTSeq statistics (v0.9.1, European Molecular Biology Laboratory, Heidelberg, Germany) to compare the Read Count values on each gene as the original expression of the gene [69,70]. The expression values were represented by fragments per kilobase transcript per million reads mapped (FPKM) [71]. Then, the difference expression of genes (DEGs) was analyzed by DESeq2 (v1.30.0, European Molecular Biology Laboratory, Heidelberg, Germany) and filtered using the threshold of |FoldChange| > 1.5 and *p*-value < 0.1 [72].

Subsequently, the upregulated and downregulated DEGs were submitted to GO and KEGG enrichment analysis [73]. Using topGO to perform GO enrichment analysis on the differential genes, calculate the *p*-value by the hypergeometric distribution method (the standard of significant enrichment is *p*-value < 0.05), and find the GO term with significantly enriched differential genes to determine the main biological functions performed by differential genes. ClusterProfiler software (v3.4.4, Jinan University, Guangzhou, China) was used to carry out the enrichment analysis of the KEGG pathway of differential genes, focusing on the significant enrichment pathway with a *p*-value < 0.05 as described previously [74,75].

### 4.5. qRT-PCR Analysis

The total RNA of each sample that was used for the aforementioned DEGs profiling was extracted by RNAprep Pure Plant Plus Kit (Tiangen, Beijing, China). Then, DNaseI treatment, RNA concentration measurement, and cDNA synthesis were carried out. According to the assembled unigenes from RNA-seq analysis, NCBI Primer-BLAST software was used to design primers for randomly selected DEGs (Appendix A). The *GAPDH* was used as an internal reference gene to calculate the relative expression of genes using the comparative Ct method [76].

### 4.6. KEGG Enrichment Analysis of DEGs and Gene Coexpression Analysis

The R package WGCNA (UC Los Angeles, Los Angeles, CA, USA) was employed to construct the coexpression networks [77,78,79] of light- and K-response based on the expression matrix of 14,465 non-redundant genes (FPKM > 1 in at least one sample, Appendix A). First, the network was constructed in accordance with the scale-free topology criterion using a soft-thresholding power β of 12 (*R*^2^ > 0.7). The coexpression modules were identified using the default settings with minor modifications, and the minimal module size and the branch merge cut height were set to 30 and 0.25, respectively. The correlation values between the coexpression modules and metabolites were calculated to determine those with high associations (|coefficient| > 0.5, *p* < 0.05).

### 4.7. Transcriptome and Metabolome Joint Analysis

The Pearson correlation coefficient was used to perform the association analysis of genes and related metabolic pathways. The relationships between metabolites and related genes were visualized by using Cytoscape software (v3.8.2, Institute for Systems Biology, Seattle, WA, USA) [80].

### 4.8. Determination of the Total Anthocyanin Content and Statistical Analysis

The total anthocyanin content of pseudobulbs of *D. officinale* was measured by UV-Vis spectrophotometry. Briefly, 0.2 g frozen pseudobulbs were treated with 1 mL of methanol:acetic acid (99:1, *v/v*) overnight in darkness at 4 °C. The absorbance was measured at 530, 620, and 650 nm, each group with six biological replicates [81].

The statistical analysis was conducted to analyze the variance among the data using the SPSS Statistics software (v22.0, SPSS Inc., Chicago, IL, USA), and the significance of the difference was detected by ANOVA followed by Duncan’s multiple range test (*p* < 0.05 as the level of significance).

## 5. Conclusions

This study deciphered the changes in the metabolome and transcriptome of *D. officinale* under the environment regulation of light and potassium. Light- and K-treatment affected the accumulation of primary and secondary metabolites in *D. officinale*, especially the biosynthesis of flavonoids. The components of flavonoid glycosides were significantly upregulated, and the accumulation of anthocyanins was the main contribution to the purple appearance of the pseudobulb. In addition, the gene modules closely related to flavonoids were obtained through WGCNA. The KEGG enrichment results of key modules showed that the accumulation of flavonoids was tightly correlated to the expression of genes related to phenylpropane, flavonoid, and the JA biosynthesis pathway. The correlation network analysis indicated a relationship among TFs, JA, and flavonoids, and JA could induce the accumulation of anthocyanins in *D. officinale.*

## Figures and Tables

**Figure 1 molecules-27-04866-f001:**
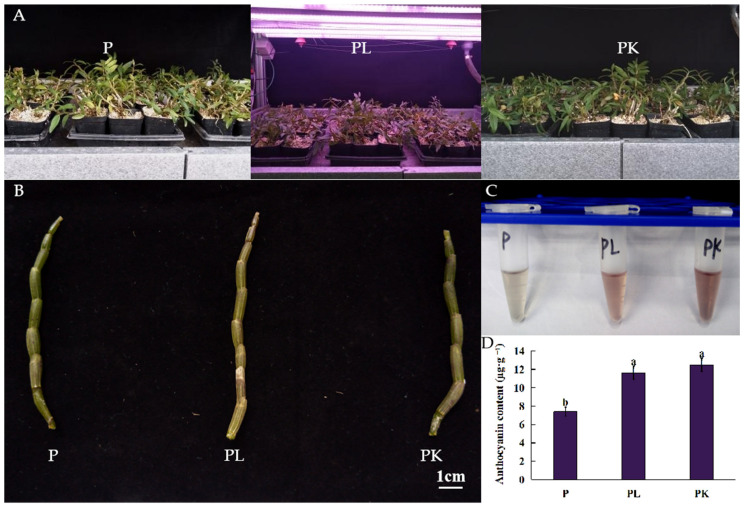
The changes in phenotype and anthocyanin content of *D. officinale* post light- or K-treatment. (**A**) The pseudobulbs (P) under natural light were used as the control, light-treated pseudobulbs (PL) were treated with red and blue light in 5:1, and K-treated pseudobulbs (PK) were treated with 3 mM KCl. (**B**) The phenotype of *D. officinale* pseudobulbs. (**C**) Anthocyanin extracted from pseudobulb of *D. officinale*. (**D**) The content of anthocyanins in *D. officinale* pseudobulbs post light- or K-treatment. The different letters represent significant difference at *p* < 0.05.

**Figure 2 molecules-27-04866-f002:**
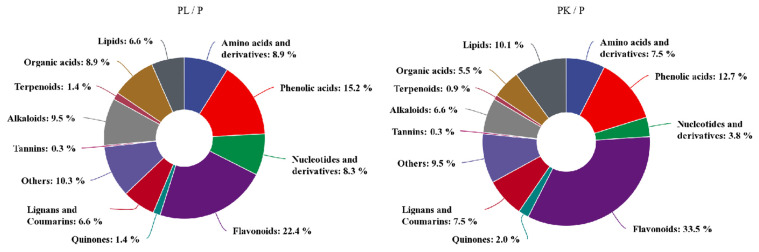
The categorization of DAMs in *D. officinale* post light- or K-treatment. PL represents light-treated *D. officinale* whereas PK means potassium-treated seedlings.

**Figure 3 molecules-27-04866-f003:**
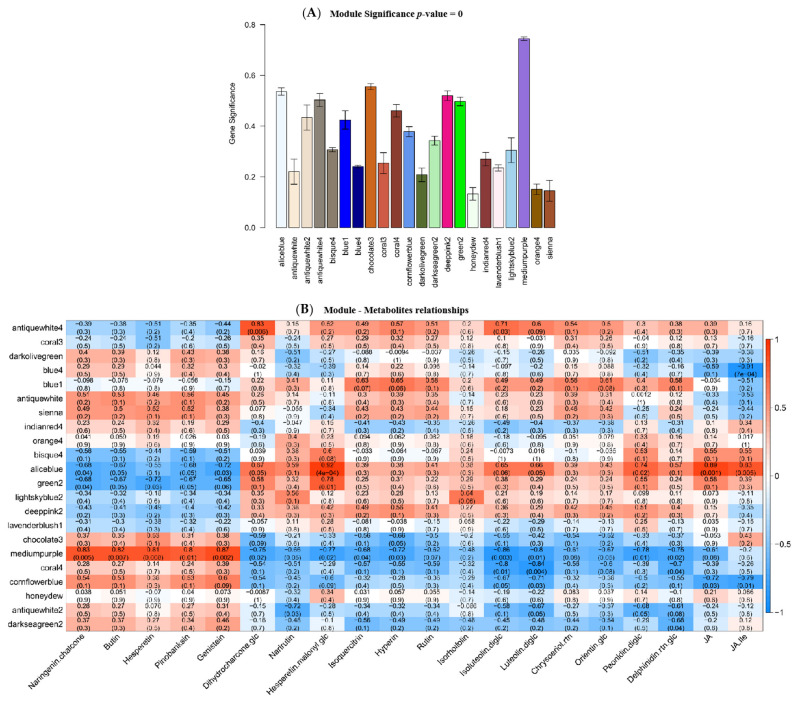
Gene coexpression analysis of light- or K-treated *D. officinale*. (**A**) The co-expressed gene modules in light- or K-treated *D. officinale* gene identified by WGCNA. (**B**) Correlation analysis of the co-expressed gene modules and metabolites. The correlation coefficient was placed in the red box, while the *p*-value is in brackets. The full information about the correlation the co-expressed gene modules and metabolites were listed in Appendix A.

**Figure 4 molecules-27-04866-f004:**
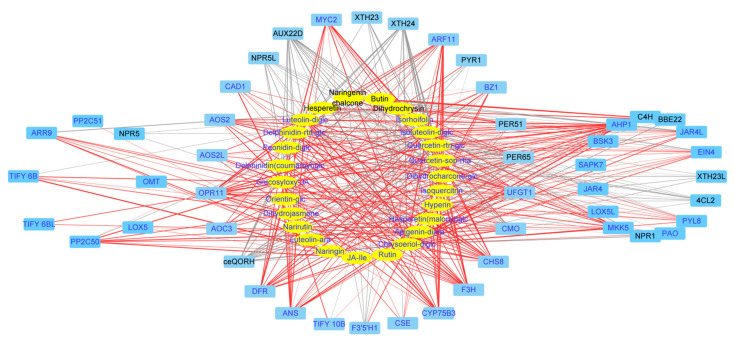
The network of DEGs in treatment-coupled modules and the altered flavonoids in light- or K-treated *D. officinale.* Metabolites are in the yellow ellipse and genes in the blue box. The upregulated DEGs or flavonoids were in blue font and those downregulated in black font. For the connection between genes and metabolites, the red lines indicated a positive correlation while the gray lines indicated a negative correlation. The treatment-altered pattern and annotation of metabolites or genes were given in Appendix A.

**Figure 5 molecules-27-04866-f005:**
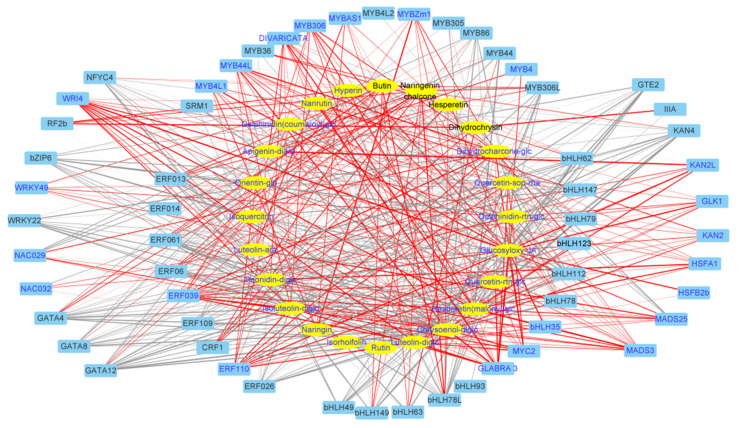
The network of differentially expressed TFs and light- and K-altered flavonoids in *D. officinale.* Metabolites are in the yellow ellipse and TFs in the blue box. The upregulated TFs or flavonoids were in blue font and those downregulated in black font. For the connection between genes and metabolites, the red lines indicated a positive correlation while the gray lines indicated a negative correlation. The treatment-altered pattern and annotation of metabolites or TFs were given in Appendix A.

**Figure 6 molecules-27-04866-f006:**
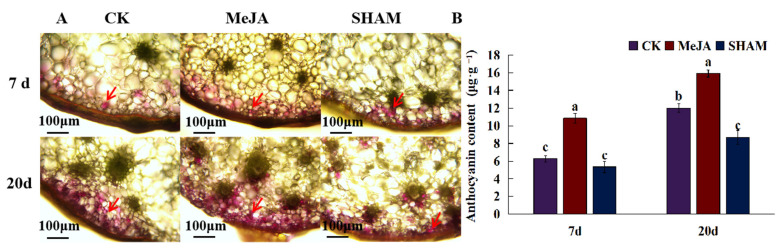
The effect of JA on the accumulation of anthocyanins in *D. officinale* pseudobulbs. (**A**) The localization of anthocyanins in *D. officinale* pseudobulbs. Red arrows indicate the accumulation of anthocyanins. (**B**) The content of anthocyanins in *D. officinale* pseudobulbs post-JA and its inhibitor (SHAM) treatment. The different letters represent significant difference at *p* < 0.05.

**Table 1 molecules-27-04866-t001:** The changed pattern of flavonoids and JAs in light- and K-treated *D. officinale*.

Compounds	CAS	Log2FC		Compounds	CAS	Log2FC	
		PL/P	PK/P			PL/P	PK/P
**Flavonoids**							
Butin	492-14-8	−0.92	−1.35	Pinobanksin	548-82-3	−0.88	−1.17
Naringenin chalcone	73,692-50-9	−0.95	−1.43	Hesperetin	520-33-2	−0.62	−1.52
Genistein	446-72-0	−0.93	−1.40	Epigallocatechin-gallate	989-51-5	−1.13	−1.66
Dihydrochrysin	480-39-7	−1.86	−1.11	(-)-Gallocatechin gallate	4233-96-9	−1.12	−2.03
Fisetin	528-48-3	−0.70	−0.67				
Narirutin	14,259-46-2	0.73	0.90	Quercetin-glc-rha	-	0.88	1.97
Naringin	10,236-47-2	0.75	0.68	Isoquercitrin	482-35-9	1.06	2.23
Isoluteolin-diglc	-	1.34	1.89	Quercetin-sop-rha	-	0.89	1.54
Diosmetin-glc	-	0.74	1.40	Hyperin	482-36-0	0.90	2.25
Isorhoifolin	552-57-8	0.65	0.89	Quercetin-rtn	147,714-62-3	0.92	2.04
Apigenin-diara	-	2.07	1.15	Tricin (sinapoyl) glc	-	0.73	0.63
Orientin-glc	-	1.15	2.12	Dihydrocharcone-glc	-	0.78	1.14
Luteolin-ara	-	0.78	0.66	Hesperetin (malonyl) glc	-	1.08	1.11
Luteolin-diglc	29,428-58-8	1.56	1.98	HydroxyKaempferol-diglc	142,674-16-6	0.98	1.16
Astilbin	29,838-67-3	0.77	0.68	Hydroxykaempferol-rtn-glc	-	0.93	1.66
Chrysoeriol-rtn	-	0.69	1.56	Hydroxykaempferol-glc	-	1.37	2.45
Chrysoeriol-diglc	-	0.61	1.02	Malonylgenistin	51,011-05-3	0.70	1.03
Quercetin-rtn-glc	-	1.12	1.77	Peonidin-diglc	47,851-83-2	0.72	0.88
Rutin	153-18-4	0.86	1.69	Delphinidin-rtn-glc	-	1.40	1.91
Quercetin-rob	52,525-35-6	0.85	1.60	Delphinidin (coumaroyl) glc	-	0.98	1.96
**JA and derivatives**							
JA	77,026-92-7	0.69	-	5′-Glucosyloxyjasmanic acid		1.83	1.35
JA-Ile	120,330-93-0	2.26	0.94	Dihydrojasmone	1128-08-1	1.44	0.73
Cis-Jasmone	488-10-8	0.72	-				

PL represents light-treated *D. officinale* whereas PK means potassium-treated seedlings. The suffix -glc is the abbreviation for -glucoside, while -diglc for -diglucoside, -ara for -arabinoside, -diara for -diarabinoside, -rtn for -rutinoside, -sop for -sophoroside, -rha for -rhamnoside, -rob for -robinobioside. CAS is the identification number for chemicals. The listed significantly differentially accumulated metabolites (DAMs) passing the threshold of |FC| > 1.5 and VIP > 0.7 with *p*-value < 0.05 post light- or K-treatment.

**Table 2 molecules-27-04866-t002:** The KEGG enrichment of DEGs in gene co-expressed modules.

GeneID	Gene	Annotation	Log2FC		*p*-Value	
			PL/P	PK/P	PL/P	PK/P
**Phenylpropanoid biosynthesis**						
LOC110097226	*4CL2*	4-coumarate-CoA ligase 2	-	−1.13	-	0.00
LOC110093998	*BBE22*	berberine bridge enzyme-like 22	-	−0.70	-	0.00
LOC110113575	*C4H*	trans-cinnamate 4-monooxygenase	-	−1.47	-	0.00
LOC110102215	*CSE*	caffeoylshikimate esterase	1.57	0.99	0.00	0.03
LOC110100929	*OMT*	Tricetin 3′,4′,5′-O-trimethyltransferase	0.62	-	0.00	-
LOC110097445	*PER51*	peroxidase 51-like	−2.55	−2.07	0.00	0.00
LOC110095178	*PER65*	peroxidase 65	−0.58	−1.22	0.02	0.00
LOC110095989	*UFGT1*	glucosyltransferase	0.77	1.29	0.04	0.00
**Flavonoid biosynthesis**						
LOC110100597	*BZ1*	anthocyanidin 3-O-glucosyltransferase	1.69	1.72	0.00	0.00
LOC110103723	*ANS*	anthocyanidin synthase	2.40	1.91	0.00	0.00
LOC110113809	*CHS8*	chalcone synthase 8	1.61	1.94	0.00	0.00
LOC110101655	*DFR*	dihydroflavonol-4-reductase	0.88	1.79	0.10	0.00
LOC110097388	*F3H*	flavanone 3-hydroxylase	1.54	1.83	0.00	0.00
LOC110113268	*CYP75B3*	flavonoid 3′-monooxygenase CYP75B3-like	1.05	1.87	0.01	0.00
LOC110103762	*F3′5′H1*	flavonoid 3′,5′-hydroxylase 1	0.94	0.88	0.05	0.00
**JA synthesis**						
LOC110093045	*AOC3*	allene oxide cyclase, chloroplastic-like	0.71	0.53	0.00	0.04
LOC110103009	*AOS2*	allene oxide synthase2	0.87	0.91	0.00	0.00
LOC110108695	*AOS2L*	allene oxide synthase 2-like	0.64	0.55	0.00	0.00
LOC110096989	*OPR11*	Putative 12-oxophytodienoate reductase 11	0.86	1.34	0.00	0.00
LOC110108000	*LOX5.1*	lipoxygenase 5.1	2.71	-	0.06	-
LOC110108011	*LOX5.2*	lipoxygenase 5.2	-	0.69	-	0.01

PL represents light-treated *D. officinale* whereas PK means potassium-treated samples. The crossbar ‘-’ means not significantly expressed in the group.

## Data Availability

The RNAseq clean data supporting the conclusions of this article are deposited in Sequence Read Archive (SRA) database in NCBI under the accession number PRJNA853073 (www.ncbi.nlm.nih.gov/bioproject/853073, which will be released upon publication).

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
