# Peer review of "Light and Potassium Improve the Quality of Dendrobium officinale through Optimizing Transcriptomic and Metabolomic Alteration"

_molecules, 2022, doi:10.3390/molecules27154866_

Round 1
Reviewer 1 Report
The study is relevant, presents consistent and interesting results. Light treatment and K affected
the accumulation of primary and secondary metabolites in D. officinale, especially the biosynthesis of flavonoids. Flavonoid accumulation was correlated with gene expression
related to the phenylpropane, flavonoid and JA biosynthesis pathway. The correlation network indicated a relationship between TFs, JA and flavonoids, and anthocyanin accumulation possibly induced by JA.
Recommend: Figure 1, place Figure 1......A ..........B.
Improve the resolution of Figure 3A and Figure 5.
Reviewer 2 Report
The authors discuss results on molecular changes following supplementation with a source of light and potassium in the cultivation of Dendrobium officinale.
My review report is as follows.
ABSTRACT
1) In the abstract, the background could be improved to better address the research problem. For instance, authors reported changes in quality of D. officinale plants considering environmental factors, however, they stated it in the past tense ("changed", "were the main..."). Does that make reference to past studies? Or does that refer to a common issue for the species? So far, I have not read the Introduction, but my question came immediately. Let this be for the author reflect on improving this part.
2) "To 15 unveil the mechanisms beneath the cultivation-aroused variation..." This expression is inadequate. Please replace by: cultivation-induced variation.
3) "Through wide-target metabolome analysis, the proportion of flavonoids in 19 differentially accumulated metabolites (DAMs) was 22.4% and 33.5% post light- and K-treatment, 20 respectively.": does that mean that you collected plant samples before treatment and after treatment? Or did you compare independent samples with a control (non-treated) and treated individuals or plots? That should be clear in the abstract.
4) "for the quality formation of this medicinal plant". I understand what you mean, but it better be written as: for quality improvement in the cultivation of this medicinal plant.
INTRODUCTION
5) "The genus Dendrobium". Needs to be italicized. Please check that thoroughly throughout the manuscript.
6) "Modern pharmacological". Suggestion: updated pharmacological.
7) Line 42: "active components". Replace by compounds.
8) When the authors declared the objectives and perspectives upon the execution of their work, they mentioned the use o LED lights. That should be important to mention in the ABSTRACT as well. Just specify that LED light.
RESULTS
9) As the manuscripts initiate with results and place methods at the end, the authors should describe better the treatments that were used. As in the abstract, I still have the question of whether the authors used the same plants for the molecular profiling, collecting samples before and after treatments. That should initiate the results sections.
- After checking the legend of figure 1 ("The pseudobulbs (P) under natural light was used as the control, Light-treated pseudo- 100 bulbs (PL) was treated with red and blue light in 5:1, and K-treated pseudobulbs (PK) was treated 101 with 3 mM KCl.), that last inquiry become more clear. However, that is also necessarywithin the descriptions of the results section. State that clear for improvement!
10) The authors described: "The anthocyanin content in D. officinale pseudobulbs increased from 7.48 to 96 11.74 and to 12.65 μg·g−1 post light- and K-treatment, respectively (Figure 1)". How many replicates support that? What is the standard deviation or standar error of the measurements? What statistical test gives suport to that? That needs to be addressed. And I recommend that the authors provide an additional graphic to Figure 1 that shoud represent the phenotypic results that were provided along with their statistical criteria for differentiation.
11) The authors also stated that: "purple spots on D. officinale pseudobulbs increased significantly". Many questions arise: how were those purple spots quantified? How many were there? What statistical criteria were used to declare that the was a significant increase. That is not declared in Material and Methods either.
12) Line 114: "and flavonoids was the most significant". Suggestion: flavonoids accounted for the highest proportion of metabolites.
13) Table 1:
a) please clarify that CAS is the identification number for chemicals. That could be added to the legend;
b) is the variation significant? Please clarify that in the Table legend as well the authors did in the results. That sustains the self-explanatory principle of tables and figures.
14) "Furthermore, 9 randomly selected DEGs were verified by 144 RT qPCR, the results showed that the expression trend of DEGs detected by RT qPCR was 145 consistent with the transcriptome sequencing data, which confirmed that the RNA-seq 146 transcriptional sequencing results were reliable (Figure S1)": I wonder if this figure should be placed within the main manuscript. Please take that into consideration. Suggestion! Rather than just a validation, important genes were studied, providing further evidence on the studied pathway.
15) Figure 3: poor resolution. Is needs to be much improved to be included.
16) Table 2: I think that the p-values should be individually placed for each differentially expressed transcript, including one more column at the end.
17) The networks need to be improved in resolution as well as the previous figures.
MATERIAL AND METHODS
18) Was the transcriptome profile also made on 3 biological replicates? Please clarify that.
19) Line 380: Replace "conjoint" by: joint
Round 2
Reviewer 2 Report
Final observations:
- In the Abstract: "After light- and K-treatment, the D. officinale pseudobulbs 18 turned purple obviously". Remove obviously.
- Figure 3 remains in low resolution. It must be improved for publication. Otherwise, author should remove written content and consider the scale of colors only for the reader to interpret the results. A supplementary table with the correlations would be the case then.
- Please check for typo errors and misspelling throughout the manuscript.